# USING ATTENTION TO WEIGHT PARTICLES IN PARTICLE FILTERS

## ABSTRACT

Particle filters are a set of algorithms for state estimation in dynamical systems. The archetypal usage of particle filters is estimating the position and orientation of a robot based on noisy sensor readings. In many situations, sensor noise is modeled to be Gaussian, where evaluating particle observations using a Gaussian probability distribution function is a reasonable way to weight particles. In this paper, we propose using attention (i.e. softmax dot product) as an alternative particle weighting function. We investigated using attention vs the traditional Gaussian weighting function in physical and temporal localization and navigation tasks, and found time performance advantages, especially when using a GPU. At the same time, we found that attention maintains comparable accuracy to the Gaussian weighting function. Code is publicly available at `github.com/anonuser-2023/project2023`.

## 1 INTRODUCTION

Particle filters are a set of algorithms for estimating states of dynamical systems, particularly useful in nonlinear or non-Gaussian state spaces. In these algorithms, one maintains a set of state hypotheses, referred to as *particles*, about the state of the system. Each particle is weighted according to how consistent it is with some observation of the system.

A typical particle filter workflow begins with initializing some number $n$ of particles $\hat{\mathbf{S}}$ in the state space $\mathbb{S}$ according to some distribution appropriate for the application, such as initializing them uniformly if one makes no assumptions about initial states. Then, a transition $t : \mathbb{S} \rightarrow \mathbb{S}$ is applied to each particle according to one's model of the system. Then, each particle $\hat{\mathbf{S}}_i$ is given a weight $\boldsymbol{w}_i, i \in \{1, 2, \ldots, n\}$ according to how consistent it is with the observed data $\mathbf{X}$, e.g. from a robot's sensors, and with the model.

In situations where one can model sensor noise as Gaussian, the traditional approach to weighting particles is to evaluate a Gaussian probability distribution function (PDF) with a standard deviation that models the sensor noise. For instance, using one-dimensional observations, if an observation is $x$, and one believes sensor noise has a standard deviation of $\sigma$, then each particle observation $\hat{\boldsymbol{x}}_i$ is initially given a weight $\boldsymbol{w}_i = \frac{1}{\sigma\sqrt{2\pi}}e^{-\frac{1}{2}\left(\frac{x-\hat{\boldsymbol{x}}_i}{\sigma}\right)^2}$. Then, to ensure that all weights sum to 1, each weight is divided by the sum of of the weights, i.e. $\boldsymbol{w}_i \leftarrow \frac{\boldsymbol{w}_i}{\sum_j \boldsymbol{w}_j} : j \in \{1, 2, \ldots, n\}$. One may also use other distributions, such as the t-distribution or the Poisson distribution, if one believes that they better model the system. Some other work has been done on alternative weighting functions, including proposing functions that output multiple weights (Zocca et al., 2022) and using neural networks (Jonschkowski et al., 2018).

Then, the highest-weighted particles are resampled, often with some perturbation added. One may then obtain a single estimate of the state by aggregating the particles, such as by taking their weighted mean $\frac{\sum_{i=1}^{n} \boldsymbol{w}_i \hat{\mathbf{S}}_i}{n}$.

**In this paper, we investigate an unconventional particle weighting method: using attention** (Bahdanau et al., 2016) **to quantify how consistent particles are with observations of the system.** Although attention was originally proposed as an add-on to machine translation techniques that were popular at the time, attention-based neural networks known as transformers (Vaswani

et al., 2023) are currently the central component of virtually all modern natural language processing (NLP). The attention mechanism assigns weights to vector representations of language tokens, called embeddings, according to their similarity with each other, calculated by taking their pair-wise dot products, and then taking the softmax of the result. That is, for a sequence of $n$ embeddings, each a vector $\boldsymbol{x}$, we have a symmetric matrix of attention weights $\boldsymbol{W}$ such that $\boldsymbol{W}_{i,j} = \boldsymbol{x}_i \cdot \boldsymbol{x}_j$. If we wish for the weights to sum to 1, then we take their softmax as $\mathrm{softmax}(\boldsymbol{W}) = \frac{e^{\boldsymbol{W}}}{\sum e^{\boldsymbol{W}}}$.

Because using attention to weight particles is so unconventional, we have not found any closely related work.

In the rest of the paper, we will introduce the characteristics of using attention to weight particles, as well as expected use cases (Section 2). Then, we investigate its behavior using a set of experiments focused on evaluating its physical localization (Section 3.1), physical navigation (Section 3.2), temporal localization and navigation (Section 3.3), and speed (Section 3.4). We then summarize our findings and propose future work (Section 4).

## 2 ATTENTION FOR WEIGHTING PARTICLES

The choice of attention is unintuitive because of an obvious weakness: it is more of a similarity measure between the direction of two vectors, whose ideal observation space consists of vectors that all have the same magnitude. In many observation spaces that occur in robotics, a common application of particle filters, such behavior is undesirable. A simple example is shown in Figure 1a, which shows the weights of 64 particles uniformly distributed between -1 and 3 according to their similarity to the scalar 1. For particles greater than 1, attention assigns weights higher than it does to particles that are closer to 1, which is undesirable, where using a Gaussian PDF to weight particles results in more desirable behavior. Attention is better suited to assessing the similarity between token embeddings in NLP tasks because embeddings tend to have magnitudes that are more similar. Figure 1b shows the similarities of 400000 100-dimensional GloVe (Pennington et al., 2014) pre-trained embeddings from Wikipedia 2014 and Gigaword 5 to an arbitrarily selected embedding. The horizontal axis is the sum of element-wise differences between each embedding and the selected embedding, another way to quantify similarity between vectors. In this case, we found using the vanilla softmax function resulted in a single peak weight at 0 and almost 0 weight elsewhere, unlikely to be useful. Thus, we introduced a sharpness parameter $\alpha$, commonly employed in smooth maximum functions such as the Boltzmann smooth max or the LogSumExp function, in order to make this weighting function more flexible. The sharpness parameterized softmax function is then $\mathrm{softmax}(\mathbf{X}, \alpha) = \frac{\alpha e^{\mathbf{X}}}{\sum \alpha e^{\mathbf{X}}}$. This effectively changes the base of the softmax function to $e^{\alpha}$. In the figure, we show $\alpha = \frac{1}{4}$.

Overall, we expect that attention may be a reasonable option for weighting particles that have relatively similar magnitudes, such as embeddings, while it may encounter problems in situations where differences in magnitude are important. Nonetheless, it may exhibit surprising robustness in such magnitude-sensitive situations, as we demonstrate in our experiments.

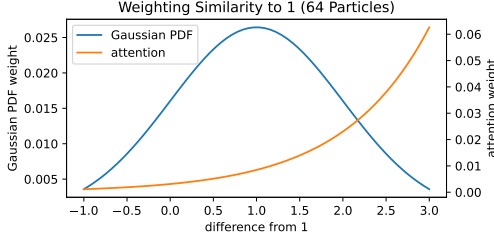
(a) Similarity between the scalar 1.

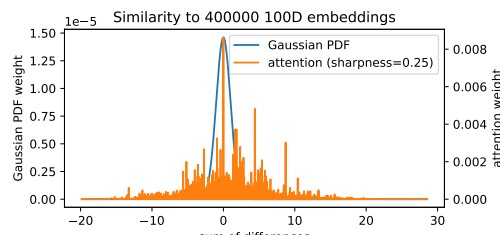
(b) Similarity between 100-dimensional embeddings.

Figure 1: When measuring similarity between vectors with widely varying magnitudes (such as scalars), attention may incorrectly weight parallel vectors too high. Vectors of overall similar magnitude can be reasonably weighted using attention.

## 3 EXPERIMENTS

### 3.1 PHYSICAL LOCALIZATION

This experiment investigates the behavior of attention as a particle weighting function when used to localize an agent using distances from landmarks. We created 2- and 3-dimensional Gymnasium (Towers et al., 2023) environments simulating continuous physical space. The environments were 100 units long on each side. The action space was any combination of movements within $[-1, 1]$ in each direction. Observations consisted of distances to each landmark with some Gaussian noise added, whose mean was 0 and standard deviation was 2. With some adjustable probability, the distance to some landmark may be perceived as not a number (NaN), which simulates random obstructions in the environment.

To compare the accuracy of the two weighting functions, we implemented an agent in PyTorch 2.0 that uses particle filters to estimate its position. Particles' observations were the expected distances to 10 randomly distributed landmarks. We started the agent at the middle of the square or cubic state space and allowed it to take random actions for 1000 time steps, updating the state estimation each time. To estimate the agent's position, we used the weighted mean of 1000 particles, which were weighted using either a Gaussian PDF or attention.

- Using the Gaussian PDF, for each particle, we summed the squared element-wise differences between the agent's observation and the particle observation, replacing NaNs with 0, and evaluated the result on a Gaussian PDF. We then divided the weights by the sum of all weights so that they sum to 1.

- Using attention, we calculated the dot product between the agent's observation and the particles' observations. We then replaced NaNs with $-\infty$ and calculated the sharpness-parameterized softmax of the result. We found the sharpness setting $\alpha = \frac{1}{8}$ to usually exhibit good results.

At each step, we dropped the 10% lowest-weighted particles and resampled with replacement from the remaining particles with Gaussian perturbation of mean 0 and standard deviation 1 until the dropped particles were replaced.

Figure 2 shows histograms of the distances between the estimated agent position and its true position using both weighting functions. Attention exhibits comparable accuracy to the Gaussian PDF across 2 and 3 dimensions and across different obstruction probabilities. When their behaviors diverge, no function is consistently superior.

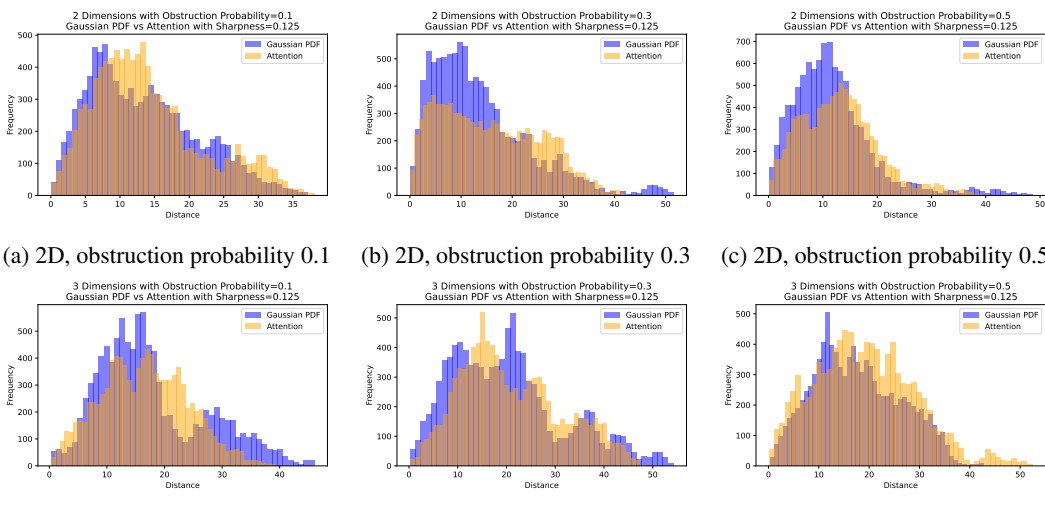

(a) 2D, obstruction probability 0.1    (b) 2D, obstruction probability 0.3    (c) 2D, obstruction probability 0.5

(d) 3D, obstruction probability 0.1    (e) 3D, obstruction probability 0.3    (f) 3D, obstruction probability 0.5

Figure 2: Histograms of differences between true position and estimated position using the Gaussian PDF vs attention weighting functions.

## 3.2 Physical Navigation

In this experiment, we used the two weighting functions to estimate the position of the agent in the environments described before, using the estimated position to move the agent to a randomly selected landmark. The localization procedure was the same as in the previous experiment. The agent's action was always the unit vector pointing in the estimated direction of the goal. We measured the probability of reaching the goal within 1000 steps and the mean number of steps taken to reach the goal if it is reached.

In most sets of environment parameters that we tried, the two weighting functions resulted in similar performance, i.e. both consistently succeeded or consistently failed. We observed contrasting behavior using 3 dimensions, 100 particles, 5 landmarks, an obstruction probability of 0.1, and a sensor noise standard deviation of 8 (8% of the environment size). Table 1 shows navigation performance using different terminal distances (the maximum distance the agent must be from the goal in order for it to be considered reached), averaged over 100 episodes. As expected, the Gaussian PDF performs better than attention in these noisy observation spaces, but we reiterate that the two functions usually perform the same and these environment parameters were chosen to contrast their behavior.

Table 1: Navigation performance averaged over 100 episodes.

| weighting function | terminal distance | success probability | steps |
|---|---|---|---|
| Gaussian PDF | 0.5 | 1.0 | 49.18 |
| Attention | 0.5 | 0.29 | 50.83 |
| Gaussian PDF | 0.25 | 0.44 | 45.09 |
| Attention | 0.25 | 0.11 | 59.73 |

## 3.3 Temporal Localization and Navigation

This experiment assesses using the two weighting functions for temporal localization of a program that approximately follows an expected schedule. Such situations may arise, for example, during parallel execution of computer programs or during the imperfect human execution of an expected series of actions. We implemented a specific instance of the latter task, namely tracking the temporal location of a musician during their performance of a musical composition with the purpose of accelerating or decelerating a prerecorded accompanying audio stream during the performance to match up with them.

For the musical composition, we used the first 26 measures of the third piano concerto[1] by Sergei Rachmaninoff (1909). We used a live recording of the concerto (Bradley-Fulgoni, 2004) and marked the times in the audio file where measures begin. (Preferably, we would obtain a recording of only the orchestra part and place more frequent markers, such as on beats instead of measures.) We then analogously marked a Musical Instrument Digital Interface (MIDI)[2] file of the piano part using notes to be played by a triangle. While estimating the soloist's position in the MIDI file during a performance, the downstream task would be to match the triangle notes in the MIDI file, played at the speed of the soloist, to the markers on the audio file by time stretching the audio file, resulting in synchronized accompaniment for the soloist.

Observations in this localization system, illustrated in Figure 3, consisted of a window of note-on MIDI events that occurred during a set number of recent time steps, represented as a matrix $X \in \mathbb{R}^{\lambda \times 128}$, where $\lambda$ is the length of the window in time steps and 128 is the number of pitches in MIDI. Although we acknowledge that other musical features such as note-off events and velocity are also relevant, we considered only note-on events for simplicity. At each time step, the last row is dropped and a new row is inserted at the beginning with ones where note-on events occurred on the

---

[1]A concerto is a kind of music that features a prominent solo instrument, in this case a piano, accompanied by an orchestra.

[2]MIDI is a way of describing musical features such as notes and the instruments that play them as a series of events. MIDI is unlike audio data which describes sound waves explicitly.

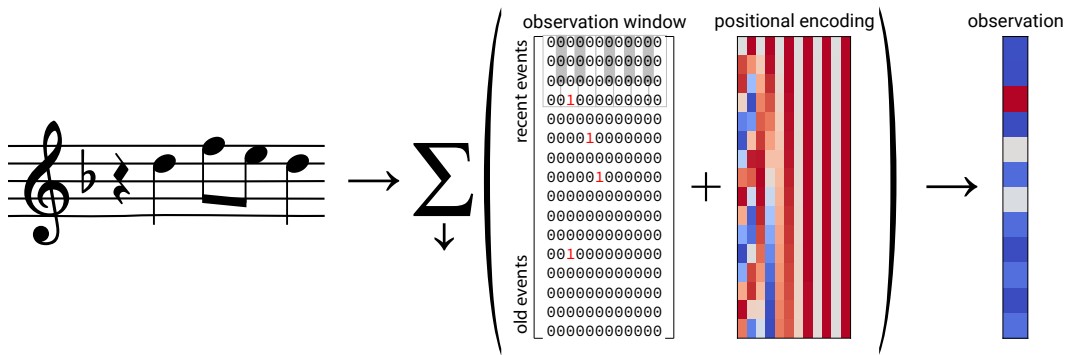

Figure 3: Interpreting MIDI note-on events as observations. We keep a rolling window of recent events, encoded as a 1 in the column of the corresponding pitch if a note-on event of that pitch occurred during that time step, with each row corresponding to each time step. We then add a positional encoding (scaled down to avoid overwhelming the event data) and sum across the first dimension. For simplicity, we show 12 pitches and an observation window of 16 time steps in the diagram. In our experiments, we used all 128 pitches in MIDI and a window of 100 time steps.

corresponding pitches and zeros elsewhere. We used a time resolution of 100 time steps per second, with a window size of 1 second. To form an observation, we add a 128-dimensional positional encoding (Vaswani et al., 2023), scaled by $\frac{1}{100}$ to avoid overwhelming the event data, to $X$. (We found that this worked better than multiplying the observations by the square root of the number of elements as in that paper.) We then summed the result across the first dimension in order to allow slight variations in event execution time from the human musician. The resulting observation is a 128-element vector. These observations are then weighted using either weighting function as described before and the weighted mean is used to estimate the temporal location of the pianist.

After estimating the pianist's location in terms of the MIDI file, the accompanying audio was accelerated or decelerated in order to match up with it (i.e. to match the audio file markers with the triangle notes), at the resolution of the markers. While there may be various ways to temporally intercept the soloist, we simply assumed that the soloist will continue going the same speed it had been going, and appropriately accelerated or decelerated the next measure to meet them. We recorded a performance of the beginning of the concerto as a MIDI file and used it to test the two particle weighting functions' accuracy in tracking the soloist in a pseudo-live performance. Tracking accuracy is shown in Figure 4. Although the Gaussian PDF weighting function tends to have lower error, both weighting functions' errors are acceptable and usually less than $10^{-5}$ seconds.

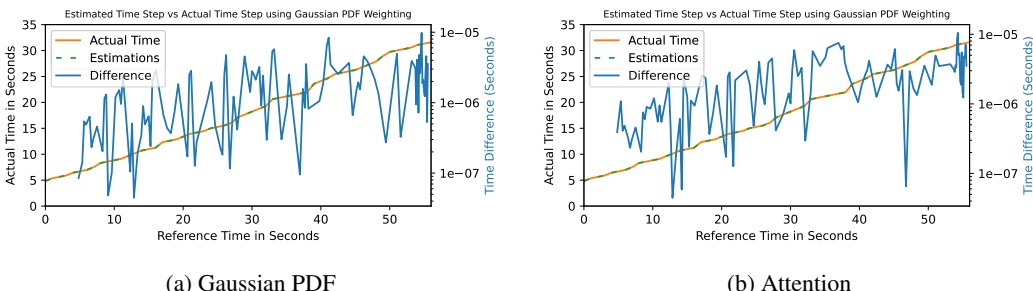

(a) Gaussian PDF

(b) Attention

Figure 4: The orange and green curves using the left vertical axis show estimated time step of the performance vs observed time step. The reference time steps are in terms of the MIDI file. The blue curve using the right vertical axis shows the error per marker. The blue curve starts only after approximately 5 seconds because the piano part does not have notes before then.

### 3.4 TIME PERFORMANCE COMPARISON

In this experiment, we measured the time taken per step of the two weighting functions in the simulated physical environments described in Section 3.1. The tests were performed on two machines.

- Machine A: AMD Ryzen 5 5600X CPU, NVIDIA RTX 3090 GPU, running Ubuntu 22.04.
- Machine B: Intel Core i5 10400 CPU, NVIDIA RTX 2060 Super GPU, running Windows 11.

Results are shown in Figure 5. As is typical of GPU-accelerated array computations, using only the CPU avoids data transfer to the GPU, which is why using only the CPU is advantageous for the small arrays that result from a low number of particles. After approximately 4096 particles, it is better to move the data to the GPU to take advantage of its higher parallel throughput. When using only the CPU, attention is faster for smaller numbers of particles while the Gaussian PDF overtakes attention at approximately 8192 particles. When using the GPU, attention outperformed the Gaussian PDF in all cases.

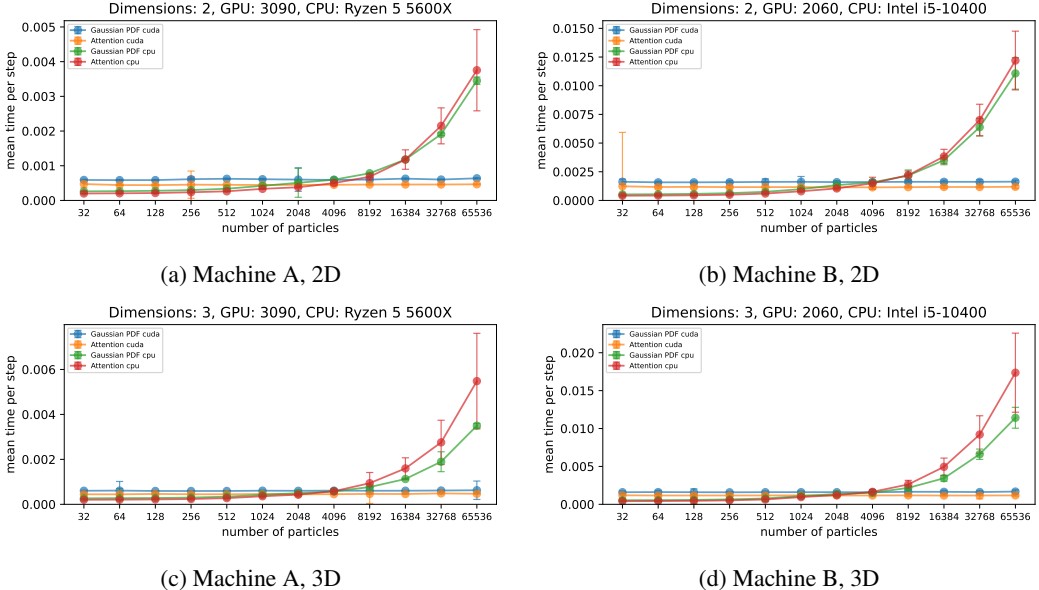

(a) Machine A, 2D          (b) Machine B, 2D

(c) Machine A, 3D          (d) Machine B, 3D

Figure 5: Mean time per step and standard deviations shown as error bars using different numbers of particles across 1000 steps. Note the logarithmic horizontal scale.

## 4 CONCLUSION AND FUTURE WORK

In this paper, we proposed using attention to weight particles in particle filters. We discussed situations in which it may be an appropriate alternative to conventional weighting functions, as well as situations in which it may fail. We experimentally evaluated its behavior in physical and temporal localization and navigation tasks, as well as its time performance compared to a conventional weighting function on different machines and using different compute devices. Through these investigations, we showed attention as an option worth considering if one is willing to sacrifice accuracy for speed, and if the application is suitable.

The temporal localization and navigation experiment was our originally intended application for this attention weighting function. For future work, we will experiment with full concertos instead of short excerpts, as well as higher-resolution labels of the audio recordings and the MIDI files. We will also consider other MIDI events in addition to note-on features, such as velocity and note-off events, as well as other ways of creating observations (e.g. weighting more recent events higher than older events), with the end goal of providing a live demonstration of a concerto performance without a human orchestra. Finally, we will investigate other applications where attention weighted particles may be considered, such as analyzing the parallel execution of computer programs.

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
