# OpenReview forum: "Using Attention to Weight Particles in Particle Filters"
_ICLR.cc/2024/Conference — ICLR 2024 Conference Withdrawn Submission_

### Official Review · Reviewer_1peL · 2023-10-30

**Soundness:** 1 poor
**Presentation:** 2 fair
**Contribution:** 1 poor
**Rating:** 3
**Confidence:** 5

**Summary:**

This is an evaluation of an alternative particle filter weighting heuristic, using dot-product similaraties (inspired by the current popularity of transformers) instead of conventional particle weights.

The approach is evaluated in the context of localization (and navigation) in a toy 3D environment, measuring accuracy and runtime, and it is also applied for synchronizing real audio tracks.

**Strengths:**

- Improving the effectiveness of particle filtering is an important problem, though the proposed method has some issues (see weaknesses).
- The exposition is generally easy to follow.
- The chosen evaluation experiments make sense, albeit somewhat limited in scope.
- A runtime evaluation is included.

**Weaknesses:**

- A related work section is missing entirely.
This is not justified from my perspective, there are plenty of works on how particle filters can be extended to neural state-space models (e.g. see [1, 2, 3]), where emissions and transitions parameterised by neural nets lead to advanced methods for both inference, and the respectively particle weighting.
To improve the paper, such methods should be discussed and the differences to them should be used to motivate the research.
- The probabilistic theory behind particle filters is not discussed in enough detail, and it is not used to justify the selected weighting scheme. Therefore, the choice to use dot-product attention as particle weights appears to be a heuristic. While there are many cases where heuristics can be useful, I could not find a precise argument in the paper for why the authors believe it would work better. The method would benefit from a more precise positioning, and ideally a better description of the mathematical fundamentals and how the method fits in their context.
- If I am interpreting them correctly, the empirical results do not show a significant advantage of the proposed particle weighting heuristic. Neither in figure 2 (which compares the results for the localization task), nor in Table 1 (which has the navigation results). The statement "the two functions usually perform the same" (comparing the Gaussian weighting baseline and the proposed method) does not match with the reported navigation success rates.
- The runtime of the method is not better than the considered Gaussian baseline on the CPU. The differences in reported runtime on GPU seem small, and might be due to GPU execution overheads (e.g. memory transfers or peculiarities of the used torch primitives). It would make sense to plot the GPU and CPU runtime results on different scales, and for the GPU many more particles should be considered.
- In terms of the 3D toy localization environment:
    - The experiments seem to assume a baseline with a Gaussian emission $\mathcal{N}(\mathbf{x}; \mathbf{z}, \sigma^2\mathbf{I})$ over observations $\mathbf{x}$ centered on a latent state (i.e. particle) $\mathbf{z}$, i.e. as if the emission model is an identity with noise $\mathbf{x} = f(\mathbf{z}) + \mathbf{e}$. But this is not what is happening in practice (based on the provided anonymized code), it should be corrected.
    - I find the term "particle observations" inaccurate, particularly because these are apparently different from the standard agent observations. I believe what was meant by this was the particles themselves, or a deterministic function thereof. I think this should be corrected.
    - The histograms in fig 2 are not normalized, which makes it harder to compare the methods.

I am sorry, but based on the overall results I cannot readily accept the claims that the proposed scheme is comparable in accuracy, but faster than conventional PF evaluations.
The paper would benefit from a more principled mathematical motivation behind the proposed weighting. It would also benefit from a thorougher evaluation that pinpoints potential advantages.

[1] Maddison, C.J., Lawson, J., Tucker, G., Heess, N., Norouzi, M., Mnih, A., Doucet, A. and Teh, Y. Filtering variational objectives. NeurIPS 2017.

[2] Le, T.A., Igl, M., Rainforth, T., Jin, T. and Wood, F. Auto-encoding sequential monte carlo. ICLR 2018.

[3] Corenflos, A., Thornton, J., Deligiannidis, G. and Doucet, A. Differentiable Particle Filtering via Entropy-Regularized Optimal Transport. ICML 2021

**Questions:**

In the provided code it seems the Gaussian weights of the localization task are based on the difference between range readings and the offsets between the particles and the known landmark coordinates, am I interpreting this correctly? If so this should be described better in the main text.

---

### Official Review · Reviewer_8Hzj · 2023-10-31

**Soundness:** 2 fair
**Presentation:** 3 good
**Contribution:** 2 fair
**Rating:** 3
**Confidence:** 4

**Summary:**

This paper proposes to use softmax dot-product attention to weight the consistency of particles in particle-filtering-based state estimation. The classical way of weighting particles is performed using a Gaussian probability distribution, in which a partition is weighted by how consistent it is with the observed data from sensors.

The idea is straightforward so the author investigated in a few physical and temporal localization and navigation tasks (physical experiment via simulation and real music localization data).

**Strengths:**

* It is rare to see the application of softmax dot-product attention in state estimation. So it is definitely a novel idea.
* The paper has a clear introduction which led to the main idea.  The paper is clearly written and easily understandable by providing intuitive examples (figure 1) and high-level figures (e.g., Figure 3).
* The paper also presents the computational comparison between the normal weighting and the attention based weighting and also found out that via GPU, the attention version actually outperformed the Gaussian PDF. So it seems the attention weighting is able to utilize the GPU (parallel computing) better.

**Weaknesses:**

* The contribution seems very limited and the application scope is widely unknown. (I would actually recommend to use a different revenue for publishing, e.g. a sub-track of the main conference)
* As the author already pointed out in section 2, attention might be more intuitive to be used as a measure for vectors with similar magnitude. This assumption is not usually valid in most state estimation scenarios.
* The sharpness parameterization the author introduced in section 2 lacks better explanation, where \alpha=1/4 seems more like a magic number and lacks theoretical support.

**Questions:**

* How do you come up with the number \alpha and the sharpness parameter? Is there any more explanation you can provide?
* How would you relax the assumption of vector magnitude (in a similar scale)? Otherwise it seems really limited in a tracking application.

---

### Official Review · Reviewer_EvVM · 2023-10-31

**Soundness:** 3 good
**Presentation:** 3 good
**Contribution:** 2 fair
**Rating:** 5
**Confidence:** 3

**Summary:**

This paper examines the possibility of using attention (borrowed from neural networks and large language models) to act as the observation probability (weight) for a particle filter, as applied to localization and navigation (in robotics and in music segments). Attention is compared to the conventional Gaussian error estimation. Results are somewhat equivocal, and the authors discuss where attention could be useful and where it will not be helpful.

**Strengths:**

- Using attention is a very interesting idea that was worth the attempt, especially given how central it is to LLMs. However, the attention has overall similar performance to Gaussian (though in quite different ways) - but to the authors' credit they seek to give insights into where attention could be useful and where it is not.

- Two quite different experiments (agent navigation, music localization) were tested, and the authors are very candid with how the systems performed.

**Weaknesses:**

- There really isn't much in it between Gaussian error and attention, in both performance and speed.

- Although the authors spend time discussing where attention may be useful, the discussion is not concrete enough to make it easy to predict ahead of time where it definitely will work well. That would be a tough task I know, but it would raise the paper's stock considerably.

- In brief, the paper is fine as a piece of research and the topic is a valid one, there just isn't enough 'weight' in the paper (results, insights, techniques) to justify recommending it for ICLR. It's a pity because I found the paper to be a good read.

**Questions:**

- Whilst the authors discuss what attention could be useful for, they tend to be rooted in the reader understanding exactly what attention is. A stronger insight or actionable method by which attention can be 'cajoled' into being a good weighting measure would be ideal, but I am not sure this is realistically achievable by the authors and/or anyone else?

---

### Official Review · Reviewer_Vkcn · 2023-11-07

**Soundness:** 1 poor
**Presentation:** 2 fair
**Contribution:** 1 poor
**Rating:** 1
**Confidence:** 4

**Summary:**

The paper proposes to use the attention computation mechanic used in transformers to compute token weights to compute weights of particles in a particle filter. Several experiments are performed to evaluate the accuracy in contrast to a simple Gaussian weighting scheme as well as the runtime performance.

**Strengths:**

From the description of the paper it is not obvious what the strengths of the proposed approaches are.

**Weaknesses:**

The description of the method feels incomplete as the text mainly gives general explanations and seemingly unrelated anecdotes. Overall the section feels like an extended introduction to the method, but is missing the crucial details. Given the simplicity of particle filters a good way to make the method clearer would be to provide pseudo code of the entire system and use that to explain the details of the proposed approach.

The resampling methodology utilized is very non-standard, at least as far as typical robotics applications go. Resampling is performed by resampling particles with a likelihood proportional to their weight, resulting in possible duplicate particles without any arbitrary noise addition. The separation of such duplicate particles is ensured through the stochastic motion model. Such a motion model appears to be missing, or at least not fully explained in the paper.

The experimental section does not demonstrate the ability of the proposed mechanism to be a suitable replacement for even a Gaussian weighting mechanism. In the navigation results shown in Table 1, the proposed method has a success rate of 29% while the Gaussian weighting has 100%. The paper states that both methods achieve similar performance, however, I'm unsure how this can be considered similar. The same holds for the second scenario. Even in  the higher dimensional task in Section 3.3 the Gaussian method has clearly lower error. Finally, the runtime numbers show that both methods seem to be roughly equal when the standard deviations are considered. If runtime wasn't reported per step but over a longer horizon the differences might be bigger or better visible.

From a robotics point of view the localization setup in the experiments is severely limited. For one it appears that only position information is considered and orientation is omitted. Second, the observations are against landmarks with a simplistic range error model. This kind of setup could also be approached with a Kalman filter which should be one of the comparisons. The scenario in which a particle filter is often employed is when using an occupancy grid map based sensor model with complex stochastic motion models. Such a setup would be significantly more convincing and possible able to demonstrate the utility of the proposed approach.

In it's current form the paper lacks the necessary detail about the method and the results do not demonstrate the benefits of the proposed method.

**Questions:**

- How does the proposed method work on more complex sensor models?